# Traumatic Vertebral Artery Stenosis Inflicted by Stray Arrow

**DOI:** 10.3390/diagnostics13071323

**Published:** 2023-04-02

**Authors:** Shan-Chia Wu, Wilson T. Lao, Chia-Hsun Lu

**Affiliations:** 1Department of Radiology, Wan Fang Hospital, Taipei Medical University, Taipei 116, Taiwan; 107148@w.tmu.edu.tw (S.-C.W.);; 2Department of Radiology, School of Medicine, College of Medicine, Taipei Medical University, Taipei 110, Taiwan

**Keywords:** stray arrow trauma, vertebral artery stenosis, arrowhead trauma, penetrating neck wound, neck injury, balloon catheter

## Abstract

A 17-year-old female presented to the emergency room with an arrow sticking out the right aspect of her neck. Her vital signs were stable with systolic blood pressure of 117 mmHg, without either tachycardia, dyspnea, or signs of active bleeding. She was fully conscious with intact sensory and motor function on all extremities. Computed Tomography (CT) showed that the tip of the arrowhead lodged at the transverse foramen of the third cervical vertebra. Digital subtraction angiography revealed that the arrowhead lies posterior to the right vertebral artery, narrowly missing it by about two millimeters. Emergency surgery was arranged in hybrid operating suite. An occlusion balloon catheter was introduced to right vertebral artery but not inflated prior to extracting the arrowhead. After extraction, oozing from the wound was noted. We then inflated the balloon while the neurosurgeon performed hemostasis with gauze compression and electrocoagulation probe. The right vertebral angiography after releasing of the balloon showed focal narrowing of the artery without contrast extravasation. The patient was discharged on the fifth hospital day, and no anticoagulant was prescribed due to lack of neurological deficit. Pre-surgical planning and partnership with the neurosurgeon lead to the optimal outcome for this case.

**Figure 1 diagnostics-13-01323-f001:**
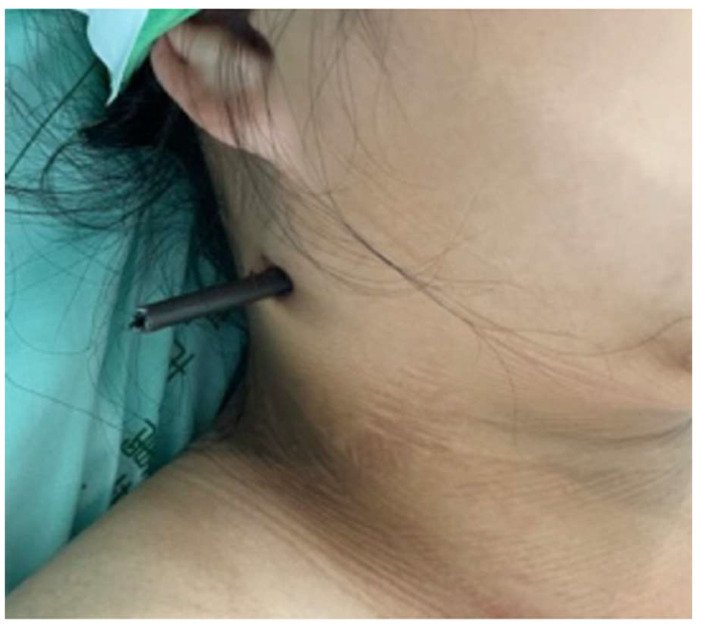
Penetrating trauma with arrow at the right side of the patient’s neck (tail end of the arrow cut and removed).

**Figure 2 diagnostics-13-01323-f002:**
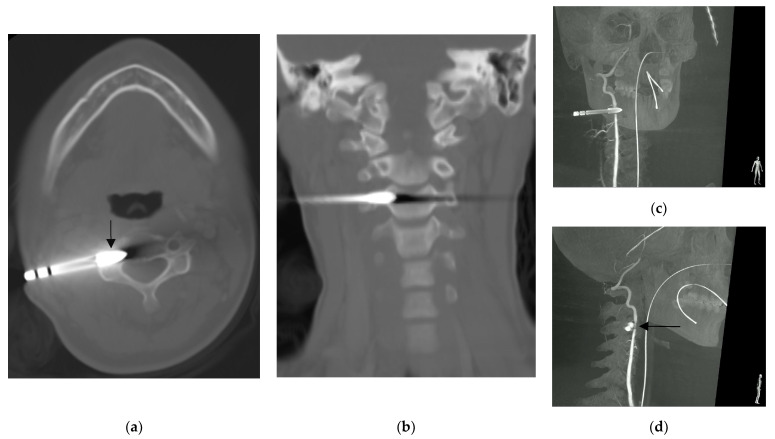
Computed tomography showing: (**a**) the arrowhead (arrow) lodged at the transverse foramen of third cervical vertebra without violating the central canal, and; (**b**) the coronal reconstruction of the same scan showed that the arrowhead did not violate the neural foramen. Right vertebral artery angiography rotation view (**c**,**d**) showed narrowing of the right vertebral artery indented by the arrowhead (arrow) [1].

**Figure 3 diagnostics-13-01323-f003:**
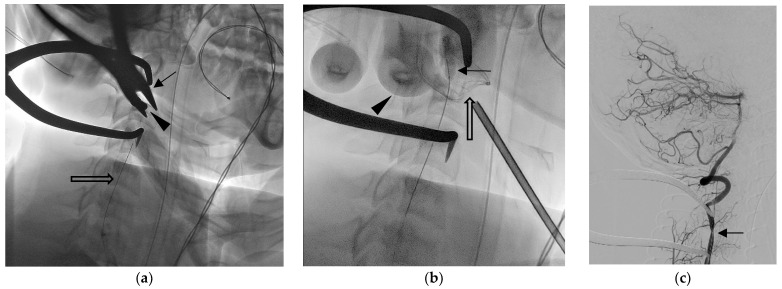
Wound exploration was performed in hybrid operating suite [2]. Fluoroscope images showed the (**a**) surgical plier (arrow) grabbing the arrowhead (arrow), and the occlusion balloon catheter was deployed but not inflated (open arrow). After arrowhead extraction, oozing from the wound was noted. (**b**) The balloon was inflated (arrow) to decrease blood flow while the neurosurgeon applied direct compression (finger imaged, arrowhead) with gauze (open arrow) before performing a further hemostasis procedure. After that, the balloon was released. The following right vertebral angiography (**c**) shows focal narrowing of the artery lumen without extravasation. The patient reported no neurological deficit either when discharged or during their visit to the outpatient clinic 3 months later [3,4,5,6].

## Data Availability

All data are available within the article.

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
