# Peer review of "Traumatic Vertebral Artery Stenosis Inflicted by Stray Arrow"

_diagnostics, 2023, doi:10.3390/diagnostics13071323_

Round 1

Reviewer 1 Report

Review Report

·     In this case report, the authors explored a rare case of traumatic vertebral artery stenosis inflicted by stray arrow.

·     The case is interesting. The authors have worked hard to present this case report in a good manner. But I have some comments:

       -Please add a brief introduction and discussion of your case.

Author Response

As requested by the Instructions for Authors of the journal, in the section for Interesting Images it was stated that: "The number of images are at the discretion of the author. No regular manuscript text (introduction/methods/results/discussion) should be included... Also, an unstructured abstract of no more than 200 words should be included..." Due to the word count limit, further discussion was not provided. And, although there was no word count limit for image legend, we thought it was inappropriate to put discussion that was not directly explaining the images in there. The intention to share this case is to concisely demonstrate the contingency we came up with to manage the potential massive bleed from the vertebral artery should it arise when the arrowhead is extracted. No similar case or management was found in our prior digital text search using Google Scholar. If any other explanation/information for the images or case deemed necessary by you, we would  provide all the needed material in our capabilities.

We deeply appreciate your comments and look forward for your further inputs.

Sincerely,

Chia-Hsun Lu

Reviewer 2 Report

This is an interesting case that is suited for presentation as an Interesting Image case in Diagnostics. The authors handled the case admirably, and I believe many readers would benefit from this short presentation, if they unfortunately find themselves with a similar case. 

That being said, I would like to ask the authors to clarify on one point. After extraction of the arrow, the authors state that "oozing from the wound was noted, injury to artery was suspected." However, the authors do not describe if there was actual injury to the artery. Please specifically describe if there was or injury to the artery or not, and also the hemostasis process that was undertaken. Furthermore, were any anticoagulants temporarily prescribed?

I understand that this will increase the length of the article. I suggest that the specific vital values be omitted.  

Author Response

Q1: The authors state that "oozing from the wound was noted, injury to artery was suspected." However, the authors do not describe if there was actual injury to the artery. Please specifically describe if there was or injury to the artery or not and also the hemostasis process that was undertaken.

Reply:

As shown in Figure 3 image (c), there was only mild narrowing of right vertebral artery lumen after arrowhead extraction, without contrast extravasation. We do not possess further evidence to prove or disprove arterial injury.

Though no photo was taken, the neurosurgeon stated that he dissected the wound reaching to arrowhead tip before extracting it. And, the hemostasis procedure was performed with gauze packing and electrocoagulation probe. It was not mentioned in image legends of the draft, but in Figure 3 image (b), the fingers of neurosurgeon and radiopaque fibers of the gauze can also be seen. After compression, suspicious bleeders were cauterized. After that, the oozing subsided. The neurosurgeon did not expose the vertebral artery from its surrounding connective tissue, which carries its own risk.

We speculated that, even if there was arterial wall injury, it probably achieved hemostasis facilitated by external compression, and internally supported by balloon expansion and slowed blood flow, which was the intention of placing the balloon in the first place, i.e. contingency should bleeding ensue after arrowhead extraction.

Q2: Were any anticoagulants temporarily prescribed?

Reply:

This was a clinical dilemma for the trauma team members. Should anticoagulants be administered for a patient who was just treated for bleeding from penetrating wound? The temporary balloon occlusion lasting about 10 to 12 minutes does poses significant risk of emboli formation. However in the end, we chose watchful waiting rather than anticoagulant usage based on the following reasons:

(1) Heparing infused saline pump was utilized throughout the procedure.

(2) No overt perfusion defect was detected during the angiography run immediately after releasing the baloon, as show in Figure 3 image (c)

(3) No neurological deficit detected either by physical exam or reported by the patient herself during visit in the post-anesthesia care unit or later in the ICU.

We deeply appreciate your comments. We will try to rephrase and add the descriptions for hemostasis device (i.e. gauze compression and electrocoagulation probe), and the reasoning to omit anticoagulant. We look forward to any further inputs from you.

Sincerely,

Chia-Hsun Lu

Round 2

Reviewer 1 Report

Thank you for your response. The case is interesting and worth publication.

Reviewer 2 Report

The authors have addressed my concerns. This is an interesting case that will be of interest to readers of Diagnostics.